# Horseradish: A Neglected and Underutilized Plant Species for Improving Human Health

**Stuart Alan Walters**

School of Agricultural Sciences, Southern Illinois University, Carbondale, IL 62901, USA; awalters@siu.edu;
Tel.: +1-618-453-3446

**Abstract:** Horseradish is a flavorful pungent herb that has been used for centuries to enhance the flavor of food, aid in digestion, and improve human health. Horseradish is a neglected and underutilized plant species (NUS), especially concerning the potential benefits to improve human health. The roots of this plant have been known for centuries to provide effective treatments for various human health disorders and has a long history of use in traditional medicine. Horseradish is a source of many biologically active compounds and its richness in phytochemicals has encouraged its recent use as a functional food. The medicinal benefits of horseradish are numerous, and this plant should be promoted more as being beneficial for human health. Glucosinolates or their breakdown products, isothiocyanates, are responsible for most of the claimed medicinal effects. Recent studies have suggested that glucosinolates provide prevention and inhibitory influences on different types of cancer, and horseradish contains high amounts of these compounds. Other medicinal benefits of horseradish include its well-known antibacterial properties that are also attributed to isothiocyanates, and its high content of other antioxidants that benefit human health. Additionally, horseradish contains enzymes that stimulate digestion, regulate bowel movement, and reduce constipation. Horseradish is a species that is vastly underexploited for its abilities as a medicinal plant species for improving human health. The health promoting effects of horseradish are numerous and should be used in an extensive marketing campaign to improve consumption habits. Consumers need to be made more aware of the tremendous health benefits of this plant, which would most likely increase consumption of this valuable NUS. Although horseradish is a highly versatile plant species and holds great potential for improving human health, this plant can also be used to enhance biodiversity in landscapes and food systems, which will also be briefly discussed.

**Keywords:** *Armoracia rusticana*; health food; human health; medicinal specialty crop; neglected species

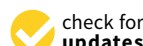



## 1. Introduction

Horseradish (*Armoracia rusticana* P. Gaertner, B. Meyer & Scherbius; *Brassicaceae*) has been cultivated for more than 3000 years for its white, thickened, and pungent roots that are generally grated and used as a condiment. This hardy, perennial herb is a large-leaved plant that forms a rosette of large, entire margined leaves having long flowering stalks with small white flowers that are borne in a terminal panicle [1]. As a member of the *Brassicaceae* family, this plant species is related to cabbage, mustard, and other cruciferous vegetables.

Although horseradish is grown commercially in many cold-temperate regions of the world, it is classified as a minor specialty crop due to its limited cultivation. The main areas of commercial production are found in Europe and North America, with other smaller production regions occurring in Asia, South Africa, and Russia. Horseradish readily escapes from cultivation and has now naturalized throughout many areas of the world, including most of Europe and central and northern North America [2]. Outside of cultivation, it is often observed in fields, home gardens, weedy areas, farmlands, roadsides, ditches, along riverbanks, and disturbed areas in eastern and northern Europe and cold-temperate sections of North America.

Considering the importance of horseradish in many cultures and the years that this plant has been in cultivation, there is a significant lack of scientific information available for this crop from limited research, making it truly a neglected and underutilized crop species (NUS). Those plant species that have some potential for human use to which little attention is paid or that are entirely ignored by agricultural researchers, plant breeders, and policymakers are NUS [3]. Although horseradish is primary known as a food additive or condiment species, it also has tremendous value as a landscape plant due to its low maintenance perennial habit and attractive foliage. Horseradish can often be observed in some rural home landscapes in Europe and North America, but it is not promoted as a home landscape plant. Thus, this review will first provide a brief historical context of horseradish, as well as culinary and landscape uses, but primarily focus on the neglected and underutilized human health benefits of this plant.

## 2. Brief Historical Context of Horseradish

Horseradish is native to southeastern Europe, and although much history of this plant remains somewhat unknown, this pungent herb has been appreciated and respected for its medicinal and gastronomic qualities for centuries. Horseradish has been used for specific purposes in various cultures for at least the last 4000 years. Oftentimes repeated non-truths become truths in history, and there is no doubt that much of the history that we associate with horseradish is probably not true. This crop was probably not known to most Egyptians in 1500 BC as much societal literature indicates, and enslaved Jews did not first find horseradish in Egypt. Horseradish is not a biblical herb nor is there evidence that the plant was used in Egypt [4]. For the Jewish Passover, horseradish has been used as a replacement for lettuce to represent a bitter herb on the Seder plate for those Jews who moved into Central and Eastern Europe. This resulted from the lack of bitter lettuce availability in this region during the early spring months in which Passover typically occurs [4]. Since most Jews living in America descended from Central and Eastern Europe immigrants, they use horseradish on the Seder plate. So, this is just an example of the complicated history of horseradish, and this review will try to focus on basic truths in a brief overview of horseradish history.

Although there is relatively little published history about horseradish, the Greeks and Romans were the first civilizations to speak extensively about the importance of this plant. Both cultures published works referring to horseradish cultivation and use for food and medicinal purposes. Since this crop has Eastern European and Russian origins [5], it has a long history in both of these regions for culinary and medical uses. During the Renaissance, horseradish consumption spread from Central Europe northward to Scandinavia and westward to England. By the late 1600s, horseradish was the standard accompaniment for beef and oysters among the British [1]. It was used primarily as a medicine by early Europeans, but later became a popular condiment to conceal the taste of tainted meats. Both roots and leaves were used as a medicine during the Middle Ages, and the root was widely used as a condiment on meats in Germany, Scandinavia, and Britain. Horseradish was introduced into the United States from Europe by early settlers in the 1600s and became popular in gardens in the New England states by the early 1800s [1].

Although horseradish is popular in certain areas of the world, especially Europe, North America, Russia, and Ukraine, and to a lesser extent in other countries, such as Australia and South Africa, this crop is still not utilized to its fullest extent. Horseradish for culinary use is a tradition of the Christian Easter and Jewish Passover in Eastern and Central Europe, and is an essential part of the traditional wedding dinner in the Bavarian region of Germany. In Austria, horseradish is embedded in the culinary culture, and can readily be found in many markets as a fresh product (Figure 1). Fresh horseradish in Austria is ground and placed on many foods from salads to cooked beans. Additionally, it is generally included on cold meat and cheese trays at Buschenschanks, which are rustic, inn-like restaurants that serve wine and cold foods. In America, horseradish is generally reserved as a vinegar-based condiment with prime rib beef or used in cocktail sauce for cold

shrimp. There are numerous products containing horseradish on American grocery market shelves, including various mustard- or mayonnaise-based sandwich spreads (Figure 2). The future of horseradish holds promise not only as culinary ingredients in various foods but more importantly as a plant species to improve human health.

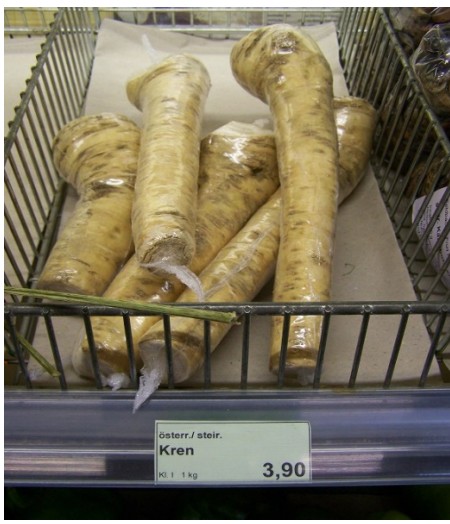

**Figure 1.** Plastic wrapped fresh horseradish available in an Austrian market (photo by Dr. Alan Walters).

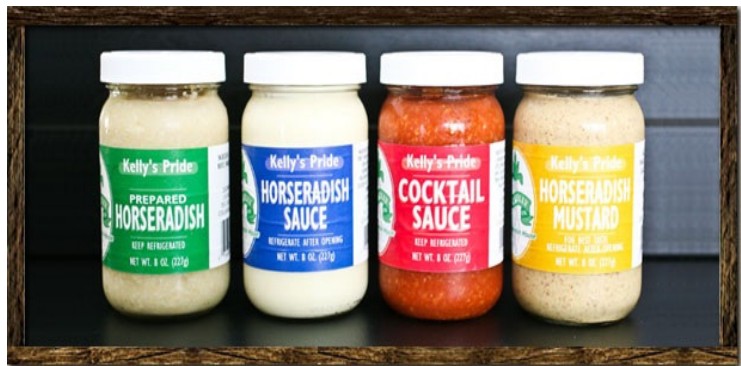

**Figure 2.** The four basic horseradish products typically found on American store shelves (left to right): prepared in vinegar, prepared in a mayonnaise-based sandwich spread, cocktail sauce for seafood, and prepared in a mustard-based sandwich spread. Photo courtesy of J.R. Kelly Co., Collinsville, IL, USA.

## 3. Biochemistry/Phytochemical Composition of Horseradish Pungency

The pungency of horseradish roots results from the sulfur-containing glucosinolates in the tissues that break down into isothiocyanates [6,7], although this plant contains many other nutraceutical compounds. The intense pungency of horseradish roots is primarily caused by isothiocyanate compounds (mostly sinigrin and 2-phenylethylglucosinolate) that result from the hydrolysis of glucosinolates by the naturally occurring enzyme myrosinase [6,8]. Today, horseradish is best known for adding a pungent flavor to all kinds of condiments that are generally used with different types of meats. Freshly grated or ground horseradish roots are highly pungent, and as the root cells are crushed, volatile compounds, known as isothiocyanates, are released. Consumer interest in improving health through the consumption of plant-derived nutraceuticals could increase demand for horseradish products due to the high amounts of these valuable health-promoting compounds contained in this plant.

Glucosinolates are nitrogen and sulfur-containing secondary plant metabolites present in all cruciferous plants [9]. All glucosinolates have a chemical structure with a sulfonated moiety, a *β*-D-thioglucose group, and a variable side chain [10]. The biological activities of glucosinolates can generally be credited to their hydrolysis products (mostly isothiocyanates), which reduce the risk of lung, stomach, colon, and rectum cancers [11]. It is thought that the isothiocyanate sulforaphane derivative of 4-methylsulfinylbutyl glucosinolate and other isothiocyanates may help prevent tumor growth by blocking the cell cycle and promoting apoptosis [10,12]. Moreover, certain glucosinolates are potential potent cancer prevention agents due to certain hydrolysis products having the ability to induce phase II detoxification enzymes, such as glutathione-S-transferase, quinone reductase, and glucuronosyl transferases [13].

Sulfur-containing glucosinolates provide the bitter flavor and pungent characteristic of horseradish, as a result of their breakdown into isothiocyanates [14–16]. Singirin accounts for about >80% of the total glucosinolates content in horseradish roots [7,15]. Once horseradish root tissues are ground or crushed, sinigrin (or other glucosinolates) mix with myrosinase, and pungent volatile allyl compounds (isothiocyanates) are produced [1,16]. Thus, glucosinolates react with myrosinase that lead to the formation of biologically active isothiocyanates, such as sulforaphane and indole-3-carobinol [1,11]. Myrosinase is the enzyme responsible for the hydrolysis of the parent glucosinolates into the biologically active products [15]. Pungency results from the glucosinolate-hydrolysis products allyl isothiocyanate and phenethyl isothiocyanate derived from sinigrin and gluconasturtiin, respectively [6].

Most of the commercial horseradish crop is crushed fresh into sauces or used as food additives for its pungent flavor [11]. Vinegar (acetic acid) is often added immediately after grinding to stop this reaction and stabilize the flavor of horseradish. Ground horseradish will slowly lose its pungency, become darkened, and develop off flavors even under proper refrigeration conditions over time (7). This quality loss can be slowed further by adding a fat or oil product, and this is the reason horseradish is often sold in combination with a product, such as mayonnaise. In comparison, freshly ground horseradish should be consumed quickly to minimize the loss of volatile pungency compounds [1].

The increased understanding and knowledge of glucosinolates and their breakdown products combined with traditional knowledge of their medicinal properties is promoting the use of horseradish roots and leaves in functional foods and medicines to act as cancer-protecting agents [7]. Horseradish is a source of many biologically active compounds and the richness in phytochemicals has encouraged its use as a functional food [17]. The increasing interest in plant-derived secondary metabolites, such as glucosinolates and other antioxidants in horseradish, provides a perfect opportunity to promote this plant as a functional nutraceutical food that can improve human health.

## 4. Horseradish Consumption and Culinary Uses

Horseradish is a flavorful pungent herb that has been used for centuries to enhance the flavor of food, as well as to aid in digestion and improve health. In the 17th century, the utilization of horseradish changed primarily from traditional medicinal and therapeutic uses to mostly culinary [18]. Both the leaves and roots of this plant are still used today to prepare many traditional dishes in many cold-temperate countries around the world. Grated horseradish roots have only two calories per teaspoon, is low in sodium, and provides potassium, calcium, magnesium, and phosphorus as well as dietary fiber, and is recommended as part of a healthy, low-fat diet because of its fat-free, high-flavor qualities [19].

Although horseradish has been prized for its medicinal and gastronomic qualities for centuries, many consider it a traditional food of the older generation, who are still attached to past traditions. Horseradish is a very traditional food for many European cultures and is widely used in Eastern European and Jewish cuisine. Throughout the world, horseradish is primarily used as a condiment in grated form or made into a paste or sauce to enhance

the flavor of other foods, especially meats. Grated or ground roots are often combined with cream, mayonnaise, mustard, oil, salt, vinegar, or yogurt. In many European countries, and in Russia, horseradish is typically freshly grated and used for salad, or made into a paste or sauce condiment [5]. Moreover, in Jewish and Slavic cultures, chrain is used to describe a spicy ground horseradish paste. The typical white chrain is grated horseradish with additions of vinegar, and sometimes salt and sugar, while red chrain includes beet roots for red coloration [20]. Although chrain is a typical condiment for various fish and meat dishes in Eastern European cuisines, this product is an essential accompaniment to Jewish gefilte fish. In comparison, horseradish sauce in North America may simply be grated roots in vinegar, or is often added to condiments, such as mustard, mayonnaise, or seafood cocktail sauce, to provide additional flavor and spiciness [21]. Vinegar stops the enzymatic reaction that provides the pungent flavor to horseradish and stabilizes the hotness of the finished product [19]. Additionally, grated horseradish roots can also be dried or powdered for various other culinary purposes.

In many areas of Europe, such as the Bavarian region of southern Germany, Kutná Hora district in the central Bohemian region of the Czech Republic, and Styrian region of southeastern Austria, horseradish is widely used in local traditional foods. In these regions, horseradish is utilized more as an ingredient to food, than as a condiment, and is very versatile and often added to desserts. Horseradish roots are still used for pickling pears in some parts of Albania, and horseradish-infused pickled pears can store safely for 6 to 8 months, with the pickled pear juice used as a hangover remedy [17]. Additionally, in many areas of Eastern Europe, horseradish is often added to fermented plant products, such as cabbage, tomato, and pepper. In southern Italy, particularly in the Basilicata region, there are many traditional culinary uses and is typically widely used as a base for preparing dishes during the carnival period [22].

In these and other areas of Europe where horseradish is part of the everyday culinary culture, it is becoming increasingly more popular in fine cuisines. However, consumption of horseradish has decreased globally in recent years and even more so since less time is spent preparing traditional meals and preserving foods for later use. The consumption of horseradish in the U.S. is relatively low, with an estimated 11,000 metric tons of horseradish roots ground and processed annually to produce approximately 22.7 megaliters of prepared horseradish [21]. This would equate to an annual consumption rate of approximately 70 mL per person in the U.S. This is a very low consumption rate considering the health benefits of this food, compared to many other foods that are consumed on a more regular basis with less human health attributes. Thus, horseradish needs to be viewed as more than just a condiment to add flavor and to compliment foods, so that consumption habits can be increased for this NUS. Consumer culinary habits need to include this herb, and advertising easy recipes in various media forms is one way to potentially increase knowledge and consumption habits. For example, horseradish added to mashed potatoes, coleslaw, or soups provides that extra sharpness of flavor to these oftentimes bland foods. Numerous horseradish products are typically available at supermarkets or specialty food stores. These include cream-style prepared horseradish, shredded horseradish, horseradish sauce, beet horseradish, and dehydrated horseradish, while many others, such as cocktail sauce, cheese, specialty mustards, and other sauces, dips, spreads, hummus, relishes, and dressings, may also contain horseradish [21].

This versatile herb has a world of opportunities as an ingredient to make more flavorful and healthy foods. It remains widely underutilized as a food additive to improve not only the flavor of foods, but also nutraceutical contents. Although horseradish roots are generally ground and consumed as a condiment, horseradish can also be consumed raw, pickled, or cooked. Most horseradish is utilized by the restaurant industry, but there are plenty of options for purchasing both fresh and prepared horseradish for home use. Fresh horseradish is generally available in produce sections of markets year-round, but the best time to purchase roots are in the spring, since roots have just been harvested and tend to be fresher with the highest firmness at this time. It is best to consume fresh horseradish within

a month after the purchase date. Fresh roots should be placed into a plastic bag in the refrigerator to maintain their freshness for the longest possible time. Once horseradish is grated or ground, it rapidly loses the pungent flavor unless placed into vinegar or a cream. Small chunks of cut horseradish roots can be frozen and stored in a freezer for up to six months, and then ground for use as needed. Prepared horseradish is most often preserved in vinegar and salt, but it is also available containing numerous other ingredients, such as mustard or mayonnaise-based spreads.

Compared to the horseradish root, culinary uses of leaves are limited, with most common uses being for salad or mixed with other herbs or vegetable species [22]. During spring months, tender leaves of horseradish were often traditionally eaten alone or mixed with other wild plant species [1]. Additionally, this plant was also sometimes used as a pot herb, with leaves boiled, water drained, and then boiled another time to eliminate the bitter of harmful substances before consumption. Horseradish leaves are still used in some countries. In Poland and Romania, leaves are placed into the bread dough and baked [5] or placed under baking bread to prevent the bread from sticking to the pan as well as to partly flavor the bread [23]. In the Basilicata region of southern Italy, horseradish roots are generally used, with leaves rarely used in food preparation [22]. Although horseradish leaves are still utilized by some cultures, their use appears to be declining, and leaves of this plant are less appreciated than roots for culinary uses.

## 5. Medical Uses of Horseradish

Horseradish has a long history as a medicinal plant, and the first use of this plant by humans was most likely for its medicinal properties, which dates back more than 3000 years. Horseradish has been known for centuries as an effective treatment for various human health disorders and has a long history of use in traditional medicine [1]. Horseradish possesses numerous therapeutic properties. Due to its valuable influence on human health, this herb has been used by many cultures throughout history to treat many human aliments, such as bronchitis, sinusitis, urinary bladder infection, paradontosis, rheumatism, anemia, gastritis, pleurisy, and to ease pain associated with sciatica and rheumatism [17]. Traditional medicinal uses of horseradish are still utilized in certain countries of eastern Europe and Russia [5]. Oftentimes to ease pain, a paste made from grated roots or leaves is placed in a cloth and applied to the skin. Additionally, fresh leaves can be applied on the surface of wounds, and for treatments of breast and skin inflammations [24]. This plant was also widely used in traditional medicine as an expectorant, to soothe respiratory issues, and help relieve rheumatism by stimulating blood flow in inflamed joints [1,12]. The suggested traditional use for colds and respiratory infections is about 20 g of fresh root per day. Horseradish syrup is used as a cough medicine due to its pungency, and grated roots are also added to various types of liquors (brandy, vodka, etc.) and later used to treat cough and bronchitis symptoms [17]. Additionally, one of the most common uses of horseradish was as a remedy for scurvy, due to its high ascorbic acid content.

Glucosinolates or their breakdown products are responsible for most of the claimed medicinal effects in horseradish. The glucosinolate, sinigrin, provides numerous benefits to the human body, including providing anticancer, anti-inflammatory, and antimicrobial properties [25]. It has also been used to speed wound healing when topically used. Besides horseradish roots being a rich source of glucosinolates, other human health constituents, including phenolics, and vitamins and minerals are also present in high amounts [17,26]. Horseradish contains high amounts of potassium, calcium, iron, magnesium, phosphorus, and potassium, with raw horseradish roots having an average of around 79 mg of vitamin C per 100 g. The average vitamin C content of horseradish roots can be almost three times higher than in citrus fruits [26]. Moreover, the complex combination of phenolic compounds contained within horseradish has antioxidant activity, which are effective inhibitors against pancreatic lipase [27].

Ancient Greeks used horseradish as a rub for lower back pain and promoted it as an aphrodisiac, while the Romans used the plant for many of those aliments previously

described. During the Middle Ages, roots and leaves were both used medicinally. Since that time, horseradish has been used as a traditional medicine, wherever it has been grown [1,7,18]. Today, horseradish has been approved in Germany for the treatment of respiratory tract infections (bronchitis, sinusitis), supportive treatment in urinary tract infections and for urinary stones, and topical use for minor muscle aches [1]. Diluted horseradish juice is also used as an antibiotic mouthwash in the treatment of periodontal disease.

Horseradish has great potential for use in the medical and pharmaceutic industries and is used in many anticancer drugs and functional foods. There are many modern therapeutic applications of horseradish, with its pharmacological benefits currently receiving great interest. The high levels of phytochemicals contained in horseradish have encouraged its use in the medical field and as a functional food [26]. In recent years, horseradish has gained tremendous interest by the scientific community and consumers, since it has been shown to contain high amounts of anticancer substances. Recent studies have indicated that horseradish provides inhibitory effects on breast, colon, lung, pancreas, prostate, and stomach cancers [27]. Moreover, horseradish has been shown to have antibacterial, antiseptic, and diaphoretic properties [1].

The medicinal benefits of horseradish are numerous, and this plant should be promoted as a plant source that is extremely beneficial to human health. Thus, several of the important medicinal benefits of horseradish are highlighted. First and foremost is the ability of glucosinolates to help prevent cancer. A high intake of cruciferous vegetables has been associated with a lower risk of lung and colorectal cancer in some epidemiological studies [28]. Glucosinolates are powerful cancer fighters, and horseradish contains 10 times more glucosinolates than broccoli, so even when consumed in small amounts, there are significant benefits to human health. However, human studies are inconsistent in linking cruciferous vegetable consumption and reducing cancer risk. This may be due to differences in cooking and preparation methods, or that some people may get greater cancer protection from these vegetables than others due to genetic differences in how glucosinolate compounds are processed in the body. Epidemiological studies indicate that human exposure to isothiocyanates and indoles through cruciferous vegetable consumption may decrease cancer risk, but the protective effects may be influenced by individual genetic variation in the metabolism and elimination of isothiocyanates from the body [28].

Another medical benefit of horseradish is that roots contain numerous antioxidants that are beneficial to human health [12,29]. Consumption of high amounts of antioxidant foods can help prevent or eliminate oxidative damage caused by free radicals. Excess production of free radicals is associated with oxidative damage to biomolecules, including lipids, proteins, and DNA, eventually leading to many chronic diseases, such as atherosclerosis, cancer, and other degenerative human diseases [29]. Additionally, some of the antioxidants found in the root are antimutagenic, meaning they protect parts of the body from mutagens that can cause permanent damage and harm to the human body [30]. Mutations are thought to lead to heart disease and several other common degenerative disorders.

Horseradish can also provide protection against various microbial pathogens. Many studies have demonstrated the antimicrobial and antibacterial capabilities of horseradish roots. The antibacterial properties of horseradish are attributed to isothiocyanates. Bacterial growth of *Pseudomonas* spp., *Escherichia coli*, *Serratia grimesii*, *Staphylococcus aureus*, and *Enterobacteriaceae* were inhibited on incubated slices of cooked roast beef that were exposed to horseradish essential oil and a distilled extract from fresh horseradish roots [31]. The beef with added horseradish restricted the growth of most bacteria to prevent spoilage. In another study, the antimicrobial activity of isothiocyanates extracted from horseradish root provided oral antimicrobial activity against six strains of facultative anaerobic bacteria: *Streptococcus mutans*, *Streptococcus sobrinus*, *Lactobacillus casei*, *Staphylococcus aureus*, *Enterococcus faecalis*, and *Aggregatibacter actinomycetemcomitans*; one strain of yeast, *Candida albicans*; and three strains of anaerobic bacteria: *Fusobacterium nucleatum*, *Prevotella nigrescens*, and *Clostridium perfringens* [32]. These results suggest that the isothiocyanates

extracted from horseradish root may be a candidate for oral use as an antimicrobial agent against microorganisms. However, the antibiotic properties of horseradish have been known for centuries, as it has been used for many years in traditional medicine to treat bronchitis, sinusitis, coughing symptoms, and the common cold [1,5]. The pungent smell of this cruciferous vegetable also helps expel mucus from the upper respiratory system to prevent infection. Again, due to the ability of glucosinolates to prevent microbial and bacterial growth, it has also been shown as an effective treatment for acute urinary tract infections. Sinigrin, which is the most abundant glucosinolate found in horseradish roots, is known to prevent water retention and act as a natural diuretic, which can help to prevent kidney and urinary tract infections. Additionally, allyl isothiocyanate, which is expelled in urine and has anti-bladder cancer abilities, may also provide infection-fighting properties in this human organ [33].

Horseradish contains enzymes that stimulate digestion, regulate bowel movements, and reduce constipation [12]. Horseradish also stimulates the production of bile in the gallbladder to aid in digestion, and bile helps rid the body of excess cholesterol, fats, and other wastes, as well as to support a healthy digestive system. Moreover, horseradish is low in calories and provides a small amount of fiber, which is very important for promoting digestive health and preventing constipation [12].

Horseradish is often applied topically to areas of the body with pain caused by injury, arthritis, or inflammation, which is due to the anti-inflammatory properties of many of the beneficial compounds it contains [1,12]. In traditional medicine, horseradish was often used to reduce pain and inflammation caused by various ailments, as well as pain associated with headaches.

Horseradish is a species that is underexploited for its vast abilities as a medicinal plant species for improving human health. Horseradish has been used for millennia as a plant species used to treat many aliments. Today, this plant is still an underappreciated medical plant species, although many of its medical attributes have been confirmed through scientific research in recent years. Particularly important is the sulfur containing-glucosinolates that have been shown to help prevent and fight cancer. However, most consumers today have little understanding of the medical attributes of this important NUS that can easily be produced in most cold temperate gardens.

## 6. Landscape and Home Garden Culture of Horseradish

Horseradish is a beautiful, multi-purpose, leafy plant for the garden and landscape in cold-temperate climates (Figure 3). This NUS can add much to a landscape due to the tremendous ornamental and edible qualities. This plant is a large-leaved, hardy, and glabrous perennial herb that grows to a height of 120 cm, with leaves often growing to lengths of 30 to 100 cm. Horseradish plants produce numerous fragrant flowers with 5- to 7-cm long upright pedicels borne on racemes that have four sepals, four petals, and six tetradynamous stamens [1]. Long stems of white, showy flowers bloom anywhere from spring to mid-summer (depending on location) to add some extra interest in the garden or landscape. In cold temperate climates, horseradish will generally require minimal care, and grow best in full sun to partial shade. Although horseradish will grow in almost any type of soil, it does require soils that are well-drained for optimal growth. The highest yields are obtained in fertile, moist soils that have good drainage with a pH between 5.5 and 7.0 [34]. Additionally, horseradish is a fairly drought tolerant plant species, relatively pest free, and tolerant to urban pollution, making it a low-maintenance plant for the garden or landscape.

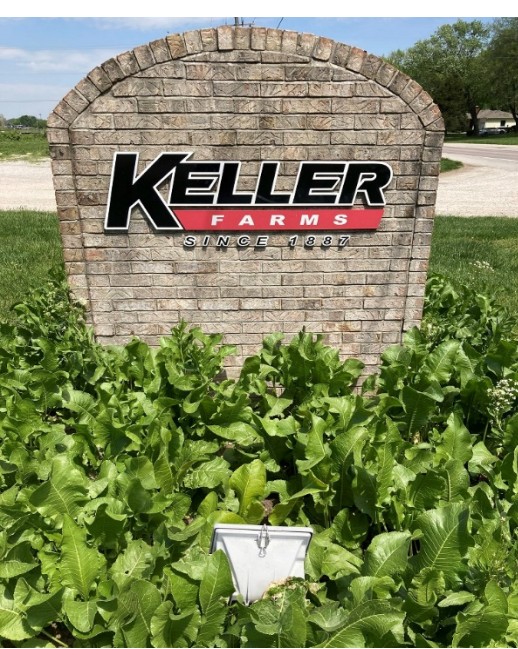

**Figure 3.** The attractive foliage and flowers of horseradish as an effective addition to a landscape (photo by Dr. Alan Walters).

Horseradish is a beautiful leafy plant that can work both as a foundation plant or focal point in a home garden or landscape. Horseradish has an upright growth habit, and some varieties can grow up to 1.5 m tall under optimal growing conditions, with large narrow leaves that typically remain dark green throughout the growing season. This plant can also be used as a focal point or as part of an herb garden. Its medium texture blends well into a garden but can always be balanced by a couple of finer- or coarser-textured plants for a more effective composition. However, it is important to note that horseradish in the garden or landscape can spread quickly, becoming invasive and difficult to remove. This ability of roots to produce new shoots allows this plant to spread prolifically and become weedy in some instances. To avoid this scenario, horseradish plants can be grown in a container or a raised bed to limit their ability to spread.

Horseradish, as plant in a garden or landscape, can also help improve overall ecological diversity. Landscapes with high plant biodiversity are more productive and stable compared to those with a low species diversity, and the loss of species diversity affects important ecosystem functions on which humans depend [35]. When many closely related plant species are grown together, the resulting plant populations produce less biomass than plant species-rich systems. Thus, low plant biodiversity results in less ecosystem productivity and supplementary plant species additions, such as horseradish, to a home or urban landscape will improve species biodiversity, leading to greater ecosystem productivity. Plant biodiversity plays an essential role to maintain healthy functioning of extensive natural landscapes that consist of different ecosystem types, including forests, meadows, or even urban areas, and recent research has also indicated that landscapes with high biodiversity can adapt better and faster to changing environmental conditions [35]. Plant species composition changes in landscapes have great influences on ecosystems when they modify those factors that directly control and respond to ecosystem processes. Plant traits related to size and growth rate are particularly important because they determine the productive capacity of vegetation and the rates of decomposition and nitrogen mineralization [36]. Thus, enriching home, rural, or urban landscapes with additional plant species that grow quickly and reach a large size at maturity, such as an NUS like horseradish, can improve the surrounding ecosystem in numerous ways. Nutrient recycling, improvement of insect and pollinator diversity, and providing food resources for bees when in flower are only a few examples of how horseradish can improve ecosystem services.

## 7. Conclusions

Although horseradish has been known since antiquity as a folk medicinal herb, natural preservative, and dish condiment [26], this plant species qualifies as an NUS due to its lack of appreciation as a valuable plant species to improve human health. Horseradish is a flavorful pungent herb that has been used for centuries to enhance the flavor of food, aid digestion, and to improve human aliments and health. Despite the highly nutritious properties of many neglected and underutilized plants, such as horseradish, they are often perceived in a negative manner, with associations of traditional food of the poor, thereby leading to rejection by some consumers [37]. However, the use of neglected crops, like horseradish, offers tremendous opportunities that place value on traditional foods with the associated culture and can also provide income opportunities. In high-income countries, the growing demand for healthier lifestyles and concern for the environment has driven a renewed interest towards sustainable agricultural systems and 'traditional' food products that can connect consumers to nature and their cultural heritage [38].

The health-promoting effects of horseradish are numerous and should be used in an extensive marketing campaign to improve consumption. Improving public communication and information about the medicinal benefits of horseradish is an essential requirement for enhancing the consumption of this neglected, underappreciated, and underutilized crop species. The use of public relations and marketing methods (e.g., advertising and media stories that spread information about the advantages of horseradish and how it can easily be used to improve the flavor and health benefit of foods) could be used to improve consumer acceptance. Consumer attractiveness of NUS products is often based on those that have numerous health benefits [39]. Another strategy that may promote and improve consumption is to use a different marketing campaign that focuses on innovative products containing horseradish, such as ice cream, potato chips, and sweet desert foods. Horseradish can bring out the flavor in even sweet dishes. Consumers need to be made more aware of the tremendous health benefits of horseradish, which would most likely increase consumption of this valuable NUS.

Thus, various marketing tools and strategies using horseradish as a "super health food" are needed to promote the key message that dietary diversity and nutritional benefits that include an NUS, such as horseradish, is a crucial component to improve human health [39]. For example, in recent years, glucosinolates have attracted the attention of the scientific community for their various medicinal and anti-carcinogenic properties, and since horseradish is an exceptional rich source of this sulfur-containing secondary metabolite, products derived from horseradish have great potential to be promoted a "super health food". Horseradish is a highly versatile plant species and holds great potential for improving human health and can also be used to enhance biodiversity in landscapes and food systems. Horseradish is truly an underestimated, underexploited, and neglected species for various reasons, but especially regarding its potential benefit to improve human health.

**Author Contributions:** S.A.W. collected all information and data relevant to manuscript content and wrote/edited the entire manuscript.

**Funding:** Not Aplicable.

**Conflicts of Interest:** The author declares no conflict of interest.

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
