# Peer review of "Horseradish: A Neglected and Underutilized Plant Species for Improving Human Health"

_horticulturae, doi:10.3390/horticulturae7070167_

Round 1
Reviewer 1 Report
MS: HORTICULTURAE-1249189
HORSERADISH: A NEGLECTED AND UNDERUTILIZED PLANT SPECIES FOR IMPROVING HUMAN HEALTH
The review report on horseradish and its uses and beneficial effects on human health.
The manuscript is generally well written and clearly presented.
I believe this review could have good value for researchers engaged in the biodiversity sector, and on neglected and underutilized plant species (NUS) in particular.
There are some points in need of improvement, highlighted in yellow and presented as note in the manuscript in the attached pdf modified file.
For this, I recommend publication after the Author have considered some minor revisions (comments).
- Pag. 1: Add email address and please check name and affiliation.
- Pag. 3: Introduction: Please highlight better paragraphs by using bold typing.
- Pag. 6: Sorry, but this words (sentences) belongs to Wikipedia??? Please check it and cite the references, improving the whole sentence.
- Pag. 8: As written before, please check and improve overall sentence.
- Pag. 8; Pag. 9 (x2): Good point, but please cite references.
- Pag. 9: Please remove hypertext links in all the review.
In the attached modified pdf file please check all the yellow highlighted sentences.

Author Response
I tried to make all the changes to the horticulturae-1249189 manuscript that you had suggested:
Page 1- I added my email address and checked my name and affiliation to make sure the were correct and I also improve the keywords
Page 3- I highlighted all paragraphs in bold type
Page 6 - I improved the sentence in question and added references
Page 8 - Again I re-wrote and improved the sentence in question
Page8; Page 9 (x2) - I added references as suggested
Page 9 and entire manuscript - I removed the hypertext links
I also corrected the highlighted areas in the manuscript that you had provided to me. Thanks for your review of this manuscript.
Reviewer 2 Report
Major review:
- The manuscript has been checked through PlagScan, which detected 17,6% level of plagiarism. Although, understandably, a review article would have a lot s of similarities with the literature, there are several sentences directly copy-pasted from other sources. Particularly problematic are the first two paragraphs on page 4 of section "Biochemistry/ Phytochemical Composition of Horseradish Pungency"
- The sentences are very long throughout the entire manuscript- and hard to understand. Please shorten the sentences and make the manuscript easier to follow and understand.
- Section Medical Uses of Horseradish: The author clearly does not have sufficient background in the medical field to correctly interpret recent findings. Also, several citations are either wrong or missing.
- 8. "Horseradish has great potential for use in the medical and pharmaceutic industries and is used in many anti-cancerous drugs and functional foods. There are many moderntherapeutic applications of horseradish, with its pharmacological benefits receiving great interest in recent years. The high levels of phytochemicals contained in horseradish have recently encouraged its use in the medical field and as a functional food [23]" Reference 23 is not correct.
- Page 8. "Recent studies have indicated that horseradish provides inhibitory effects on different types of cancer including stomach, colon, and prostate [25], as well as prevention of lung cancer and development of liver tumors [26]" No such finding is cited literature, please change the literature.
- Page 9. References are missing at half of the page.
Minor points:
- Several words in the manuscript are in different font and size. Please correct.
- Page 3. Reference missing after the sentence "In America, horseradish is widely used since most Jews living there descended from Central and Eastern Europe immigrants, who used horseradish on the Seder plate."
- Page 4. "…root cells are crushed, volatile oils, known as isothiocyanates, are release". Please rephrase volatile oil into volatile compounds.
- Page 4. What is "sulfurous characteristic ". Please explain.
- Page 5. "In this reaction, glucosinolates (such as sinigrin, released from the vacuole) react with myrosinase, causing the formation of biologically active isothiocyanates, such as sulforaphane and indole-3-carobinol". In this sentence, a reader can conclude that sulforaphane or indole-3-carobinol can be formed from sinigrin, which is not correct. Please rephrase.
- Page 5. Please correct gluconasturtin into gluconasturtiin.
- Page 5. Although "…to inhibit different types of cancer or gastric lesions" is part of the sentence from the literature, the phrase is not evidence-based so please rephrase it into "acts as cancer-protecting agents" or some similar expression.
- Page 7. "… including slowing the spread of cancer, lessening inflammation, antibiotic agent (especially against E. coli bacteria), antioxidant". Only laboratory testing was made, so please write incorrect terms such as "anti-cancer, anti-inflammatory, antibacterial, antifungal, antioxidant activity" etc.
- Page 8. ". First and foremost is the ability of glucosinolates to prevent and fight cancer." Please rephrase in "help in prevention"
- Page 8. These glucosinolates found in cruciferous vegetables "turn on" genes that suppress tumors". Tumor suppressor gene do not suppress tumors?! Please rephrase.
- Page 8. "…horseradish roots contain a number of phytochemicals, which are antioxidants". Please rephrase, it looks like all phytochemicals are antioxidants.
- Page 8. "Consuming high amounts of antioxidant-rich foods can help eliminate or prevent this damage". Which damage?
- Page 9. "Singrin, which is the most important glucosinolate found in horseradish roots". Please replace important with abundant. Also, there is "i" missing in sinigrin.
- Page 9. "Besides being low in calories and high in fiber, horseradish is also thought to help manage weight loss due to the high content of isothiocyanates" Please explain. Reference missing.
Author Response
I had provided my responses to you comments regarding the review of horticulturae-1249189 manuscript.
Major points:
- I have went those first two paragraphs in the "Biochemistry/Phytochemical Composition of Horseradish Pungency" and re-wrote those paragraphs
- I went through the entire manuscript and shortened numerous sentences, so that the reader good better follow
- I went back through the manuscript a couple times to make sure that all references were correct and I added a few more
- Changed that paragraph a bit and corrected the reference that you had pointed out
- Corrected that paragraph as you had suggested, that paragraph is now at the top of page 7.
- I added more references into that section
Major points:
- Corrected the issue in the manuscript with differing font and size.
- I changed the sentences here and added a reference, but I and most Americans know that most Jews in America descended from Central and Eastern European immigrants.
- Rephrased volatile oil to volatile compounds
- Removed sulfurous characteristic from manuscript, but that just means the smell of sulfur
- Changed the sentence, per your suggestion to rephrase to reduce confusion
- Corrected to gluconasturtiin
- Per your suggestion I reworded these sentences, see third paragraph on page 4
- Made the changes as you had suggested, see 3rd paragraph on page 6
- Rephrased per your suggestion, see page 7 paragraph 2
- Removed statement regarding tumor suppression
- Rephrased the statement about antioxidants to reduce that confusion, see page 7 paragraph 3
- Added oxidative damage
- Replaced important with abundant and corrected spelling of sinigrin, see first paragraph of page 8
- I just removed the statement regarding weight loss, as it did not add that much anyway.
Hopefully, I tried to answer and correct all your review comments regarding this manuscript. Thanks for your comments and it does make it a much better manuscript.
Round 2
Reviewer 2 Report
The manuscript is largely improved and can be accepted as it is.
Just noticed "u" missing in glcosionolates on page 3.; the second paragraph after the ref. [9].